# CanStoc: A Hybrid Stochastic–GCM System for Monthly, Seasonal and Interannual Predictions

**Shaun Lovejoy** [1] and **Lenin Del Rio Amador** [2,*]

1 Department of Physics, McGill University, Montreal, QC H3A 2T8, Canada; lovejoy@physics.mcgill.ca
2 Département de Mathématiques, Université du Québec à Montréal, Montreal, QC H2X 3Y7, Canada
* Correspondence: del_rio_amador.lenin@uqam.ca

**Abstract:** Beyond their deterministic predictability limits of ≈10 days and 6 months, the atmosphere and ocean become effectively stochastic. This has led to the development of stochastic models specifically for this macroweather regime. A particularly promising approach is based on the Fractional Energy Balance Equation (FEBE), an update of the classical Budyko–Sellers energy balance approach. The FEBE has scaling symmetries that imply long memories, and these are exploited in the Stochastic Seasonal and Interannual Prediction System (StocSIPS). Whereas classical long-range forecast systems are initial value problems based on spatial information, StocSIPS is a past value problem based on (long) series at each pixel. We show how to combine StocSIPS with a classical coupled GCM system (CanSIPS) into a hybrid system (CanStoc), the skill of which is better than either. We show that for one-month lead times, CanStoc's skill is particularly enhanced over either CanSIPS or StocSIPS, whereas for 2–3-month lead times, CanSIPS provides little extra skill. As expected, the CanStoc skill is higher over ocean than over land with some seasonal dependence. From the classical point of view, CanStoc could be regarded as a post-processing technique. From the stochastic point of view, CanStoc could be regarded as a way of harnessing extra skill at the submonthly scales in which StocSIPS is not expected to apply.

**Keywords:** long-range forecasts; stochastic forecasts; scaling; weather; macroweather; hybrid models

## 1. Introduction

The lifetime of planetary structures in the atmosphere and ocean are roughly ten days and six months, corresponding to the average energy rate densities of, respectively, $\varepsilon \approx 1$ mW/kg and ≈10 nW/kg [1]. Since the lifetimes are close to the corresponding predictability limits, at longer times (in the "macroweather" regime, [2,3]), deterministic General Circulation Models (GCMs) , become stochastic and this occurs independently of whether or not stochastic parametrization is used to enhance the small scale variability [4–10].

The ten-day and six-month predictability limits are somewhat variable from location to location [1] but they are sufficiently distinct such that there exists two somewhat different stochastic regimes; the first, a transitional regime over scales of several weeks to a year, the second, over the lower frequencies. These correspond to GCMs optimized for "long-range" (macroweather) monthly, seasonal interannual forecasts (Long Range Forecasts, LRF) and those optimized for longer-term (e.g., multidecadal) climate projections, i.e., forecasts based on scenario forcings. Whereas the main challenge of LRFs is predicting the response to (stochastic) internal forcings, the longer-term climate models attempt to average out these responses to estimate the deterministic response to anthropogenic forcings. This paper will focus on LRF models, the forecasts of which benefit somewhat from the (coupled) ocean part that still has some deterministic skill.

After a decade or more of fairly intensive effort in many countries, there are now fourteen Global Producing Centres for LRF's (GPC-LRF) that regularly send forecasts to the

WMO (https://community.wmo.int/en/global-producing-centres-long-range-forecasts, accessed on 21 November 2023). Tracing back to [11] and developing in parallel with GCM LRFs, there have also been stochastic LRFs developed, notably by [12–15], under the rubric "Linear Inverse Modelling" (LIM) models. LIM models are based on systems of (conventional) integer-ordered stochastic, not fractional, differential equations. Consequently, LIM impulse response (Green's) functions are composed of (short memory) exponentials; the memory in the system—which is the basis of the stochastic forecast—decays rapidly, its memory is "short". In comparison, starting with [16–18], stochastic LRF models (the ScaLing Macroweather Model, SLIMM) were developed based on long-range memory processes called "fractional Gaussian noises" (fGn). fGn processes are themselves based on fractional (ordered) differential equations and, hence, power law (not exponential) Green's functions. Consequently they have much stronger "long-range" memories. However, in order to exploit these long memories for forecasting, long data series are needed. Contrary to GCM models that are classical *initial* value PDE solvers, SLIMM models effectively solve a *past value* problem, the forecast skill of which improves the more past data that are used.

Initially, the use of long memory fGn processes was justified by the underlying spatial (and hence temporal) scaling of the governing equations and GCM models. However, starting with [19,20], it was justified as the high frequency limit of the Fractional Energy Balance Equation (FEBE), itself a phenomenological equation that results by applying the scaling principle to the energy storage processes. Finally, refs. [21,22] showed that an important special case of the FEBE—the Half-order Energy Balance Equation (HEBE)—follows as a necessary consequence of eliminating a key approximation in the derivation of the classical Budyko–Sellers-type energy balance models. The half-order surface temperature equation is in fact a direct consequence of the classical heat equation with the (correct) radiative–conductive surface boundary conditions. This HEBE derivation shows that the source of the long memory is in the heat conduction into the atmosphere and subsurface (ocean or land) where heat is stored, and, that this is generally a long memory, power law process. If, instead of starting with the classical heat equation, we proceed from its generalization, the fractional heat equation, then we obtain the more general FEBE.

Just as GCMs can be optimized for the LRF or lower frequency climate projection regimes, the same is true with the FEBE; SLIMM simply corresponds to the high-frequency FEBE limit with internal (stochastic) forcing. To make climate projections, one uses the full FEBE but with deterministic (external, mostly anthropogenic) forcing. Following the precursor Scaling Climate Response Function (SCRF) projections [23], ref. [24] showed how the FEBE may be used with Bayesian parameter estimates to make global temperature projections through to the year 2100, which had significantly lower uncertainties than the Climate Model Intercomparison Project GCMs used in the IPCC's fifth and sixth assessment reports (the CMIP5 and CMIP6 GCMs). Also significantly, it was also shown that "hybrid" projections made by combining the FEBE and GCM projections could do better still [25]. In this paper, we propose an analogous hybrid LRF model.

To make a complete, practical forecast system even over seasonal time scales, it is necessary to forecast the responses to anthropogenic forcings. This turns out to be easy because the SLIMM model is a linear stochastic model, so that the response to anthropogenic forcing can be forecast separately and linearly combined with the SLIMM forecast of the internal variability. The use of a linear stochastic equation such as the FEBE may be justified because of the exceptionally low intermittency in the macroweather regime (e.g., [26]). This is in contrast to the (turbulent) intermittent weather regime that requires nonlinear stochastic (cascade) models and would involve multifractal predictions.

The system that forecasts the responses to both anthropogenic and internal forcings is called the "Stochastic Seasonal and Interannual Prediction System" (StocSIPS, see: [27]). StocSIPS was first developed for global temperature forecasts in [18], and then for regional forecasts in [28,29]. Its regional forecasts were produced by independently forecasting each pixel from the pixel's time series. At first sight, it might appear that, by discarding spatial correlations (notably, teleconnections), the forecasts would not be optimal. However,

in [30], it was further shown that—contrary to intuition—neither distant (nor neighbouring) pixels had any Granger causality so that—by definition—using their information would not improve the forecasts, implying that the StocSIPS regional forecast was indeed optimal. Further, in [30], it was shown how the long memory stochastic model that underlies StocSIPS produces El Nino events. Just as in GCMs, StocSIPS does not put in the El Nino events "by hand", in both cases, they are rather a emergent features of quite different model types.

The latter paper marked the first application of Granger causality to long memory geoprocesses, and it is important in understanding climate "networks" (e.g., [31–33]). Teleconnection-inspired forecast models often use climate (especially El Niño) indices [34,35] to make regional stochastic forecasts. However, ref. [30] showed that, if the time series at each location was long enough and was correctly exploited, the spatial correlations that they imply are not useful for forecasts. The reason for this is that, in predictions using extremely long series, information from the distant past—including the influence of events such as El Nino—have already been "used". Equivalently, this analysis showed that, if regional forecasts are made with only short series of past data, then they can partially make up for the missing (distant past) data by "borrowing" recent information from other spatial locations. However, if long enough data exists, other pixels will no longer have any new information to "borrow".

Thus, there exist two complementary LRF model types: initial value GCMs and the past value StocSIPS model. How do they compare? In a recent paper, ref. [36] reviewed and evaluated the seasonal temperature forecasts of six LRF models, and also considered seven different ways of combining them as copredictors into unique optimal seasonal LRF products. The forecast skill was quantified in various ways including the standard Mean Square Skill Score (MSSS = $1 - <E^2>/<T^2>$); $T$ is the temperature anomaly, $E$ is the forecast error, MSSS = 1 is a perfect score, and forecasts with MSSS = 0 have no skill, "< ">" means ensemble average. For seasonal forecasts, ref. [36] found that, even when the optimal seasonal LRF product was used, the skill was still low, with MSSS $\approx$ 0.1 over land, and MSSS $\approx$ 0.1–0.8 over ocean with a global average MSSS $\approx$ 0.2 (depending slightly on the season, see the figures in Section 2.3). Using the same reanalyses to define the anomalies and for the evaluation, ref. [28] showed that for seasonal forecasts at zero lead times, StocSIPS generally showed comparable MSSS skill and, in the case of the temporal correlation coefficient, it showed somewhat better skill. This is not so surprising when it is remembered that, theoretically, most of the deterministic skill of the GCMs is in the first two weeks or so, whereas most of the StocSIPS skill is over time scales of two weeks and longer.

The MSSS error metric was used because, for Gaussian processes, it is the quantity that is maximized in order to yield the conditional ensemble forecast. It is also standard in the literature. Using other metrics such as the root mean square error or correlation coefficients do not change the qualitative results discussed below. See [18] for a detailed theoretical discussion and empirical analysis in the case of StocSIPS.

The thorough analyses by [36] showed that—unsurprisingly—the six individual LRF models were not very different from each other and—also unsurprisingly—that combining them to give an optimal predictor did not markedly increase the overall skill. However, StocSIPS is a past value problem: its' skill comes purely from the past. This is in complete contrast with the GCMs whose skill comes completely from the spatial structure at the initial instant. Their qualitatively different natures can therefore potentially be combined to produce a significantly better hybrid LRF. In this paper, we show how to make such a hybrid by combining StocSIPS with the Canadian Stochastic and Interannual Prediction System (CanSIPS) GCM-based LRF. The resulting product "CanStoc" is indeed a significantly improved product, especially over time scales of one month, in which their respective scales of validity overlap the most. From the point of view of conventional LRF models, CanStoc could be considered as a (greatly) improved post-processing procedure to reduce the bias of the model [37–40]. More information is available in the thesis [29].

## 2. The Models

### 2.1. StocSIPS

For readers unfamiliar with StocSIPS, in this section, we give a succinct summary; the full details are given in [18] (global), and in [28] (regional; see also [30]). A key point to remember, is that, theoretically, StocSIPS is only expected to apply to the macroweather regime, i.e., beyond the lifetime of planetary structures, which is typically around ten days (although the lifetime varies quite a bit from place to place). Since the horizontal (but not vertical) weather regime wind statistics follow the Kolmogorov law up to planetary scales (for reviews, see [1,41,42]), the lifetime of planetary structures is close to the atmosphere's deterministic predictability limits, such that the beginning of StocSIPS's prediction skill coincides with the extreme limit of deterministic (GCM) skill (this is a simplification since the ocean deterministic limits are longer—see below). The two models are thus complementary.

The development of StocSIPS was originally motivated by the empirical finding that macroweather temperatures showed scaling over wide temporal scale ranges. In addition, unlike the turbulent weather regime, macroweather temperatures had low intermittency, such that the statistics were not so far from Gaussian. Although the scaling exponents vary somewhat with geographical location, they are typically in the range of fractional Gaussian noise (fGn) processes [43], such that the latter could be used as models [16,17]. This discovery was exciting, because, unlike classical autoregressive or moving average processes, the "memories" of which decay exponentially, fGn processes have long (power law) memories, implying that the current state of the atmosphere is affected by events many time steps in the past. Indeed, as discussed above, past values are so important that forecasting fGn processes is essentially a "past value" problem (more details shortly).

At the same time, the development of scaling climate projection models involved the empirical determination of the optimal scaling exponent required for longer (multi-decadal) global climate projections [23,44]. It was soon realized [20] that this long time scale deterministic exponent and the shorter time scale stochastic exponent [16,45] could both be explained as, respectively, the long and short time scaling behaviours of a single wide-scale range model. The model—the Fractional Energy Balance Equation, FEBE—was a fractional generalization of the Budyko–Sellers energy balance model and it predicted both low- and high-frequency scaling exponents as consequences of a unique fractional derivative of the order $h \approx 0.4 \pm 0.1$. Finally, refs. [21,22] analytically derived the $h = 1/2$ special case of the FEBE by updating the Budyko–Sellers model to include the correct radiative–conductive surface boundary conditions (the Half-order EBE or HEBE). Not only is the HEBE exponent compatible with observations, but it is attractive, because it represents the theoretical solution of the classical heat equation when forced by (realistic) radiative–conductive surface boundary conditions. This means that its parameters (relaxation times and diffusion distances) have direct physical meanings and interpretations. At the same time, it unifies the high- and low-frequency responses to both stochastic (internal) and deterministic (external) forcings.

Appealing to the energy balance principle is a theoretically satisfying basis for predictions and projections, because it shows how nonclassical (long) memories arise naturally even in otherwise classical equations (here, the heat equation). The resulting (horizontal) space–time ($\underline{r}, t$) FEBE model for (surface) temperature anomalies $T$ is

$$\tau^h \frac{\partial^h T}{\partial t^h} + T = sF$$
$$\frac{\partial^h T}{\partial t^h} = \frac{1}{\Gamma(1-h)} \int_{-\infty}^{t} (t-p)^{-h} T'(p) dp \tag{1}$$

In the top equation, $F$ is the forcing anomaly, $s$ is the climate sensitivity parameter (the inverse of the cloud feedback parameter), $\tau$ is the relaxation time, and $h$ is the FEBE exponent. $T$ and $F$ are time-dependent, but the parameters $\tau$ and $s$ are only functions of position. $\Gamma$ is the standard gamma function. On the left, the response to the forcing has two contributions: the linearized blackbody radiation response (the $T$ term), and the subsurface

and atmospheric storage is the fractional derivative term defined in the second line. The latter is an order $h$ fractional derivative (it is partial—"$\partial$"—since $T$ is a function of both time and position), $T'$ is the usual derivative, and $p$ is a dummy variable of integration. A third contribution to the response comes from the divergence of the (anomalous) horizontal heat flux, but this is unimportant for long-range forecasting, and, for simplicity, we have ignored it here (see, however, the discussion in the conclusions). We already mentioned that $h = 1/2$ yields the HEBE [21], and, when $h = 1$, we instead obtain the (globally averaged) Budkyo-Sellers Energy Balance Equation (EBE) (equivalent to a "box" model). The more general case $h \neq \frac{1}{2}, \neq 1$ is the Fractional EBE (FEBE).

We could note that various definitions of fractional derivatives and integrals exist; here, we use a particularly simple one: the Weyl fractional derivative, the Fourier transform ("*F.T.*") of which is simply $\partial^h/\partial t^h \overset{F.T.}{\to} (i\omega)^h$, where $\omega$ is the Fourier conjugate variable, the frequency. When $h$ is an integer, this relationship is classical.

Since the FEBE is linear, it may be solved by impulse response (Green's) functions $G_h$:

$$T = sG_h * F; \quad (G_h * F)(t) = \int_{-\infty}^{t} G_h(t - p)F(p)dp \tag{2}$$

where "*" indicates time convolution and $s$ is only a function of position. For the classical $h = 1$ value, we have

$$G_1(t) = \tfrac{1}{\tau}e^{-t/\tau}; \quad h = 1 \tag{3}$$

and for $0 < h < 1$, the leading order terms of the small and large $t$ Green's function expansions, are, respectively,

$$G_h(t) \approx \begin{array}{l} -\frac{1}{\tau\Gamma(-h)}\left(\frac{t}{\tau}\right)^{-h-1}; \quad t > \tau \\ \frac{1}{\tau\Gamma(h)}\left(\frac{t}{\tau}\right)^{h-1}; \quad t < \tau \end{array} \tag{4}$$

(the full Green's functions are based on "generalized exponentials", Mittag–Leffler functions, see ref. [44] for a summary). Comparing Equations (3) and (4), we see that the standard integer case ($h = 1$) is very special: the impulse responses fall off very rapidly (exponentially) rather than slowly (as a power law with an exponent $-h - 1$; with $h$ not far from the value 1/2).

When $F$ is a deterministic forcing (e.g., the anthropogenic climate forcing), then the low frequency term with $h \approx 0.4$ can be used for global climate projections through to the year 2100 [24]. When the same (AR6) forcings are used, the median FEBE projections are quite close to the IPCC AR6 Multi Model Ensemble (MME) projections; the agreement between the models gives support to both. However, an important advantage of FEBE projections is that they have much smaller uncertainties. For example, the 90% MME confidence range for the AR6 climate sensitivity is 2.0–5.5 C/CO$_2$ doubling, whereas, for the FEBE, it is 1.5–2.2 C/CO$_2$ doubling. Ref. [24] explained the slightly lower FEBE sensitivity by their finding that the (negative) aerosol forcing was somewhat too strong, hence that the overall anthropogenic forcing used in the GCMs was too weak. The lower FEBE sensitivity was a consequence of using the slightly higher forcing that resulted from moderating the aerosol cooling. The issue of the correct level of aerosol forcing is still not clear, but, in any case, it will not much affect the FEBE uncertainty about the median.

In contrast to these low frequency projections, StocSIPS uses the high-frequency approximation (Equation (4), the $t < \tau$ case) to make monthly, seasonal, and annual forecasts. Over these time scales, the responses to anthropogenic forcings are smaller than the responses to the (stochastic) internal forcings and—due to the linearity—the two are forecast separately and combined at the end (see Figure 1 for an illustration). For the stochastic part, $F$ is taken to be Gaussian white noise. When this high frequency approximation is used, the response is exactly a fractional Gaussian noise (fGn) process.

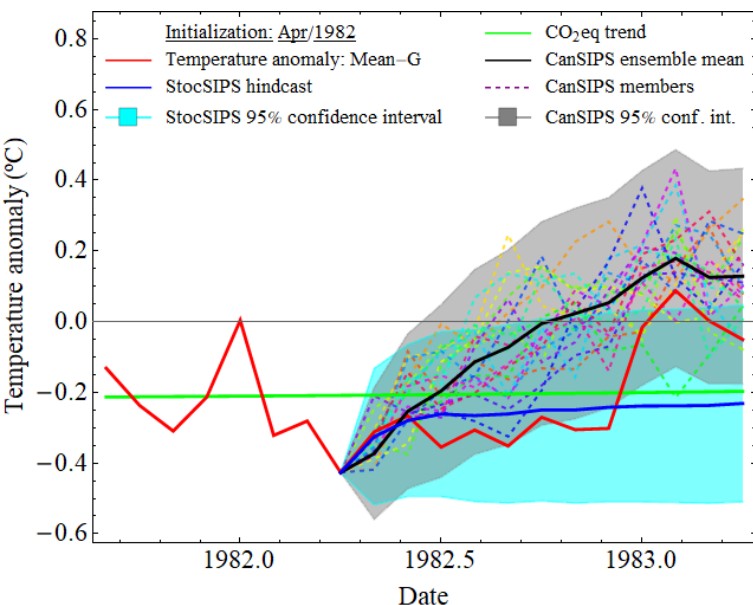

**Figure 1.** Examples of forecasts. One example of forecast for the 12 months following April 1982 for both StocSIPS and CanSIPS. In red, we show the verification curve of observations for the Mean-G dataset. In blue, the median hindcasts for StocSIPS, with the corresponding 95% confidence interval based on the root mean square error (RMSE) for the verification period. The ensemble mean for CanSIPS is shown in black, with each of the 20 members shown in dashed light colors and the 95% confidence interval based on the RMSE of the hindcasts represented in grey. The $CO_{eq}$ trend for the Mean-G dataset (green line) was added as a reference of the long-term equilibrium of the temperature fluctuations. Reproduced from [18].

fGn processes are noteworthy because of their long memories. To help understand this, consider the autocorrelation functions $R(\Delta t) = \langle T(t)T(t - \Delta t)\rangle$ ("< >" indicates ensemble average), when $F = \gamma(t)$ is a white noise (i.e., $T = sG_h * \gamma$):

$$\begin{aligned} R_1(\Delta t) &\propto e^{-\Delta t/\tau} & h &= 1 \\ R_h(\Delta t) &\propto \Delta t^{2h-1} & 0 &< h < 1/2 \end{aligned} \qquad (5)$$

(the bottom equation is for the fGn approximation, i.e., $G_h(t) = (t/\tau)^{h-1}/(\tau\Gamma(h))$), at small $\Delta t$, there is a divergence that is cutoff by the finite resolution of the process; here, by the inner scale of the macroweather regime $\approx 10$ days). We note the very slow decay in the autocorrelation for $0 < h < \frac{1}{2}$, indicating that values in the distant past affect the present value. If one defines the "integral time scale" of a process by the integral of its autocorrelation function, then, for $\frac{1}{2} \geq h > 0$, $R_h(\Delta t)$ diverges, hence these fGn processes are termed "long memory" processes.

The extent of the memory is more explicitly demonstrated when we consider the optimum fGn forecasting algorithm, which is the basic StocSIPS algorithm. Consider a discrete time series (i.e., in integer time) with data from $-t$ to $t = 0$ (the present). The forecast that minimizes the mean square error is given by a linear combination of past data:

$$\hat{T}(k) = \sum_{j=-t}^{0} \phi_{t,j}(k)T_j \qquad (6)$$

where $\hat{T}(k)$ is the $k$ steps predictor and $\phi_{t,j}(k)$ is the prediction kernel. Since the process is Gaussian, it turns out that $\hat{T}(k)$ is also the conditional expectation of the process conditioned on the observed past values $t$ time steps in the past.

In the standard $h = 1$ (box model, EBE) case, the weights $\phi_{t,j}(k)$ fall off very quickly as we move further into the past, such that only one or two terms in the sum are needed (e.g.,

$j = -1, 0$), and the process can be modelled as a (short memory) first-order autoregressive AR(1) process. However, when $<\frac{1}{2}$, the situation is quite different. Figure 2 shows the typical kernels (weights) for the various $h$ values weights (Equation (6)) of the discrete time fGn process discussed in [46]. At first (small lags, $j$), as in the $h = 1$ case, the weights fall off quickly (although as power laws) and are not so large at medium lags. However, what is new and interesting, is that, for weights far enough in the past, they become more and more important and are singular as $j$ approaches $-t$! In the continuous time case studied by Gripenberg and Norros, the divergence at the present and the distant past are both singularities of order $-h$. In the words of Gripenberg and Norros [47], "this divergence when we approach $-t$ is because the closest witnesses to the unobserved past have special weight".

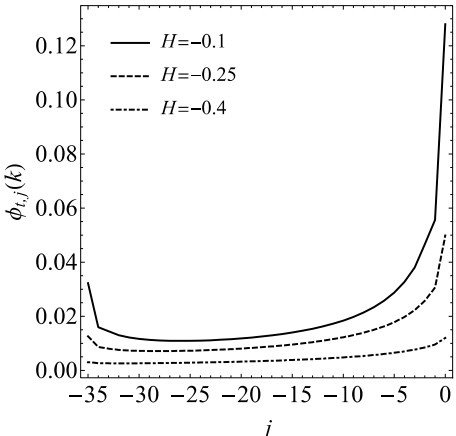

**Figure 2.** Optimal coefficients, $\phi_{t,j}$, in Equation (6) with $H = h - \frac{1}{2} = -0.1, -0.25, -0.4$, (top to bottom i.e. $h = 0.4, 0.25, 0.1$) for predicting $k = 12$ steps in the future by using the data for $j = -35, 0$ in the past. Notice the strong weighting on both the most recent (right) and the most ancient available data (left, both are singular) and how the memory effect decreases with the value of $H$ (hence, with $h$, the maximum is when $H = 0$, $h = \frac{1}{2}$). Compare to Figure 3.1 in [47]. Reproduced from [18].

The amount of memory needed to make a forecast depends on the value of $H = h -1/2$. Figure 3 shows this by plotting the minimum memory needed, $m_{95\%}$, to obtain more than 95% of the asymptotic skill (corresponding to $m = \infty$) as a function of the forecast horizon, $k$, for different values of $H$. The line $m = 15\,k$ was also included for reference. The closer to zero the value of the exponent, $H$ (the closer $h$ is to $\frac{1}{2}$), the less memory we need in order to approach the maximum possible skill. This fact seems counterintuitive, but the explanation is simple: for larger values of $H$ (closer to zero), the influence of values farther in the past is stronger, but at the same time, more information on those values is included in the recent past, so less memory is needed for forecasting. Following the rule of thumb found by [47] for the continuous case: "one should predict (. . .) the next second with the latest second, the next minute with the latest minute, etc." Actually, from Figure 3 we can conclude that, for predicting $k$ steps into the future, a memory $m = 15\,k$ would be a safe minimum value for achieving almost the maximum possible skill for any value of $h$ in the range $(0, \frac{1}{2})$, which is the case for temperature and many other atmospheric fields. Of course, if $h$ is close to $\frac{1}{2}$, a much smaller value could be taken. The approximate ratio $m_{95\%}/k$ for each $H = h - \frac{1}{2}$ was included at the top of the respective curve. From the point of view of the availability of data for the predictions, this result is important. Once the value for $h$ is estimated, assuming it remains stable in the future, we only need a few recent datapoints to forecast the future temperature. The information of the unknown data from the distant past is automatically considered by the model.

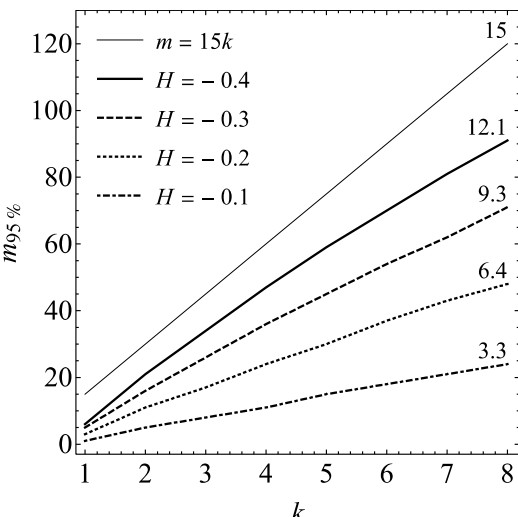

**Figure 3.** Minimum memory, *m*, needed to obtain more than 95% of the asymptotic value (corresponding to $m = \infty$) as a function of the horizon, *k*, for different values of $H = h - \frac{1}{2}$. The larger the value of *H* (the closer *h* to $\frac{1}{2}$) the less memory is needed for a given horizon. The approximate ratio $m_{95\%}/k$ for each *H* was included at the top of the respective curve. Reproduced from [18].

In Figure 1, we show an example of a forecast for the 12 months following April 1982 for both StocSIPS and CanSIPS (the Canadian GCM system described in Section 2.2, below). In red, we show the verification curve of the observations for the dataset (see [18] for full details). In blue, we show the median hindcasts for StocSIPS, with the corresponding 95% confidence interval based on the root mean square error (RMSE) for the verification period. The ensemble mean for CanSIPS is shown in black, with each of its 20 members shown in dashed light colors and the 95% confidence interval based on the RMSE of the hindcasts is represented in grey. The $CO_{2eq}$ trend for the dataset (green line) was added as a reference of the long-term equilibrium of the temperature fluctuations. As expected, the dispersion of the different ensemble members for the dynamical model increases as the horizon increases, which shows the stochastic-like character of the GCMs for long-term predictions with the consequent loss in skill. The ensemble spread score (ESS), a measure of the forecast "reliability", (see, e.g., [40]), is defined as the ratio between the temporal mean of the intra-ensemble variance and the mean square error between the ensemble mean and the observations. For CanSIPS, it is in the range 0.57–0.74 for all lead times, except for zero months lead time, in which case ESS = 0.40 [38]. Notice that the cloud of dashed lines is narrower than the grey region in Figure 1. The low ESS implies that CanSIPS is underdispersive for all horizons, whereas (by construction), StocSIPS has ESS ≈ 1 and is thus nearly perfectly reliable [18].

### 2.2. CanSIPS

The model that we used as a StocSIPS co-predictor is the Canadian Seasonal to Interannual Prediction System (CanSIPS, [48,49]) that was developed by the Meteorological Service of Canada (MSC). In this work, we used outputs from the second version of the model (CanSIPSv2, https://climate-scenarios.canada.ca/?page=cansips-technical-notes, accessed on 23 November 2023). The following details correspond to this new version.

CanSIPS is a multi-model ensemble (MME) system using 10 members from each of two climate models (CanCM4i and GEM-NEMO) developed by the Canadian Centre for Climate Modelling and Analysis (CCCma) for a total ensemble size of 20 realizations. It is a fully coupled atmosphere–ocean–ice–land prediction system relying on operational data assimilation for the initial state of the atmosphere, sea surface temperature, and sea ice.

To evaluate forecasts and compare StocSIPS with CanSIPS, we accessed the publicly available series of CanSIPS hindcasts covering the period 1981–2010 (CanSIPS 2021). The

fields, available on 145 × 73 latitude–longitude grids at resolutions of 2.5° × 2.5° for each of the 20 ensemble members, were area-weighted and were averaged to obtain the global mean series of hindcasts at the monthly resolution. CanSIPS produces a forecast at the beginning of every month for the average value of that month and the next eleven months, i.e., for lead times from 0 to 11 months for each initialization date. In our case, this corresponds to forecast horizons (number of periods ahead that are forecasted) from 1 to 12 months. In the verification for $k = 1$ month (lead zero), the hindcast period is January 1981–December 2010; for $k = 2$ months (lead one), the hindcast period is February 1981–January 2011, and so on. This way, all the 12 series of the hindcasts (one for each horizon) have a length of 360 months.

*2.3. CanStoc*

Over the relevant monthly, seasonal, and annual time scales, the response to anthropogenic warming is small and relatively easy to forecast (see [28] for details). Below, we consider the problem of predicting the stochastic response to internal forcings plus the anthropogenic response, i.e., the full anomalies with respect to the seasonal climatology. If we denote predictors by a circonflex and use the subscripts *C*, *S*, and *CS* for CanSIPS, StocSIPS, and CanStoc, respectively, then the combined (linear) "CanStoc" copredictor is

$$\hat{T}_{CS} = \alpha\hat{T}_S + \beta\hat{T}_C \tag{7}$$

where $\alpha$ and $\beta$ are weights that depend on the geographical location and on the forecast lead time. The forecast errors are thus

$$\begin{aligned} E_S(t) &= T(t) - \hat{T}_S(t) \\ E_C(t) &= T(t) - \hat{T}_C(t) \\ E_{CS}(t) &= T(t) - \hat{T}_{CS}(t) \end{aligned} \tag{8}$$

where $T(t)$ is the actual temperature. The condition in which the forecast is optimal (in the root mean square sense, e.g., [50]) is given by the orthogonality principle, which states that the error and the predictor are orthogonal (uncorrelated): $\langle \hat{T}_{CS} E_{CS} \rangle = 0$. See Figure 4 for a geometric interpretation and Figures 5 and 6 for examples of forecasts.

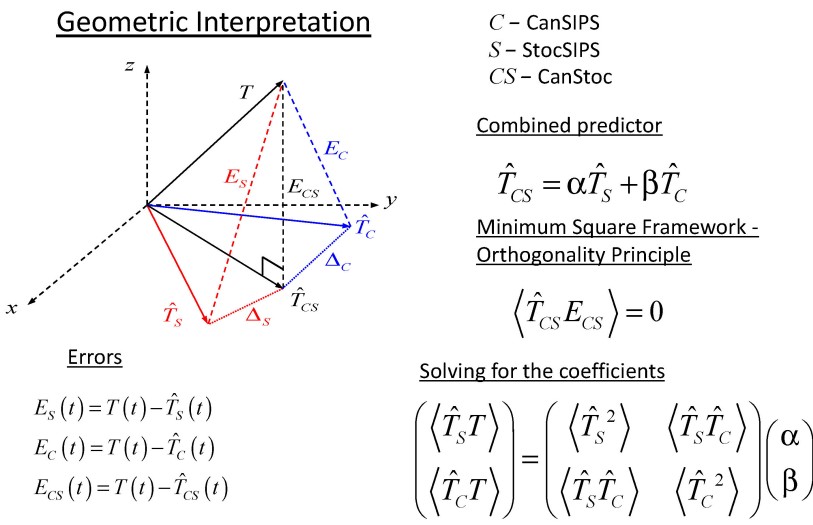

**Figure 4.** A schematic showing the construction of the hybrid forecast $\hat{T}_{CS}$ using the StocSIPS and CanSIPS forecasts as linear copredictors. The relative weights ($\alpha$ and $\beta$) are selected so that the hybrid error $E_{CS}$ is orthogonal to the predictor (hence minimizing the mean square error). A geometric interpretation is also given that shows directly how the (hybrid) copredictor (**black**) improves over the individual predictors (**red**, **blue**).

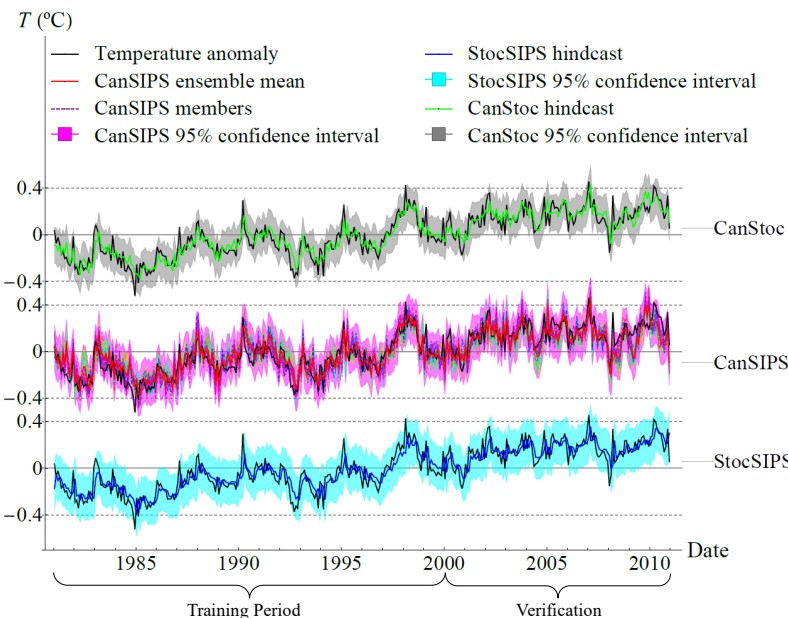

**Figure 5.** Series of hindcasts one month ahead with 90% confidence limits. The StocSIPS and CanStoc parameters were estimated over the 20-year training period indicated, whereas the verification was performed over the following 10 years.

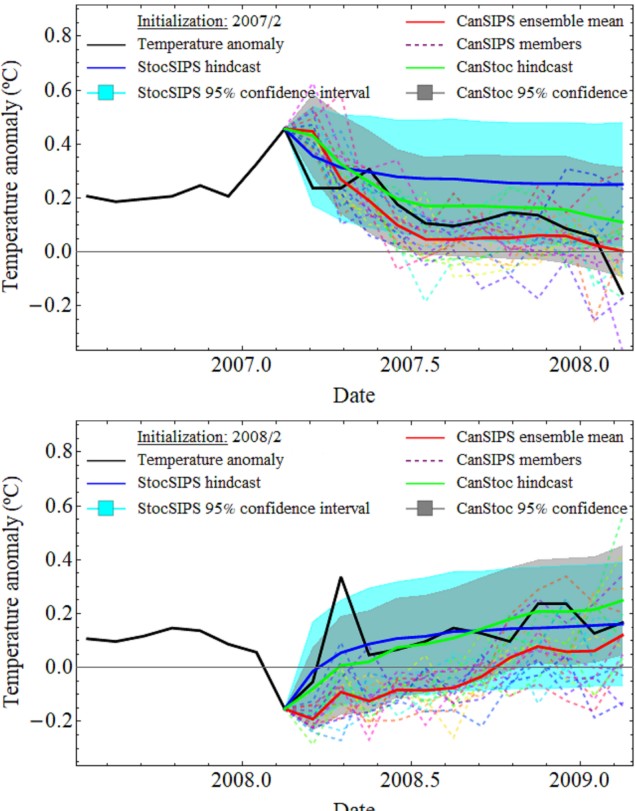

**Figure 6.** Examples of hindcasts made in February 2007 (**top**) and February 2008 (**bottom**). The actual temperature anomaly is in black, the StocSIPS mean hindcast is solid blue (95% confidence limits in light blue), with the CanSIPS hindcast in red with the individual members dashed, and the hybrid CanStoc mean hindcast in solid green with grey, indicating the 95% confidence limits. In the top hindcast, we see that CanSIPS (**red**) did somewhat better than StoSIPS (**blue**), but that CanStoc (**green**) was better still. In the bottom, StocSIPS is excellent, but CanStoc is even better.

Using StocSIPS and CanSIPS as copredictors, at each pixel, we can easily solve for the weights, α and β. Figure 7 shows the results for the lead times of 1–3 months and (right hand column) the difference between the weights. To compute the coefficients and skill scores for CanStoc, we used out-of-sample crossvalidation with 500 random partitions of 20 years for training and the complementary 10 years for validation. The rows (top to bottom), show how the weights evolve as the forecast lead times increase from 0 to 2 months. We see that, at zero months (top row), most of weight (and hence skill) is contributed by CanSIPS (most regions are blue), but, at two months lead time, over almost all land regions and much of the ocean, StocSIPS makes the largest contribution to the skill.

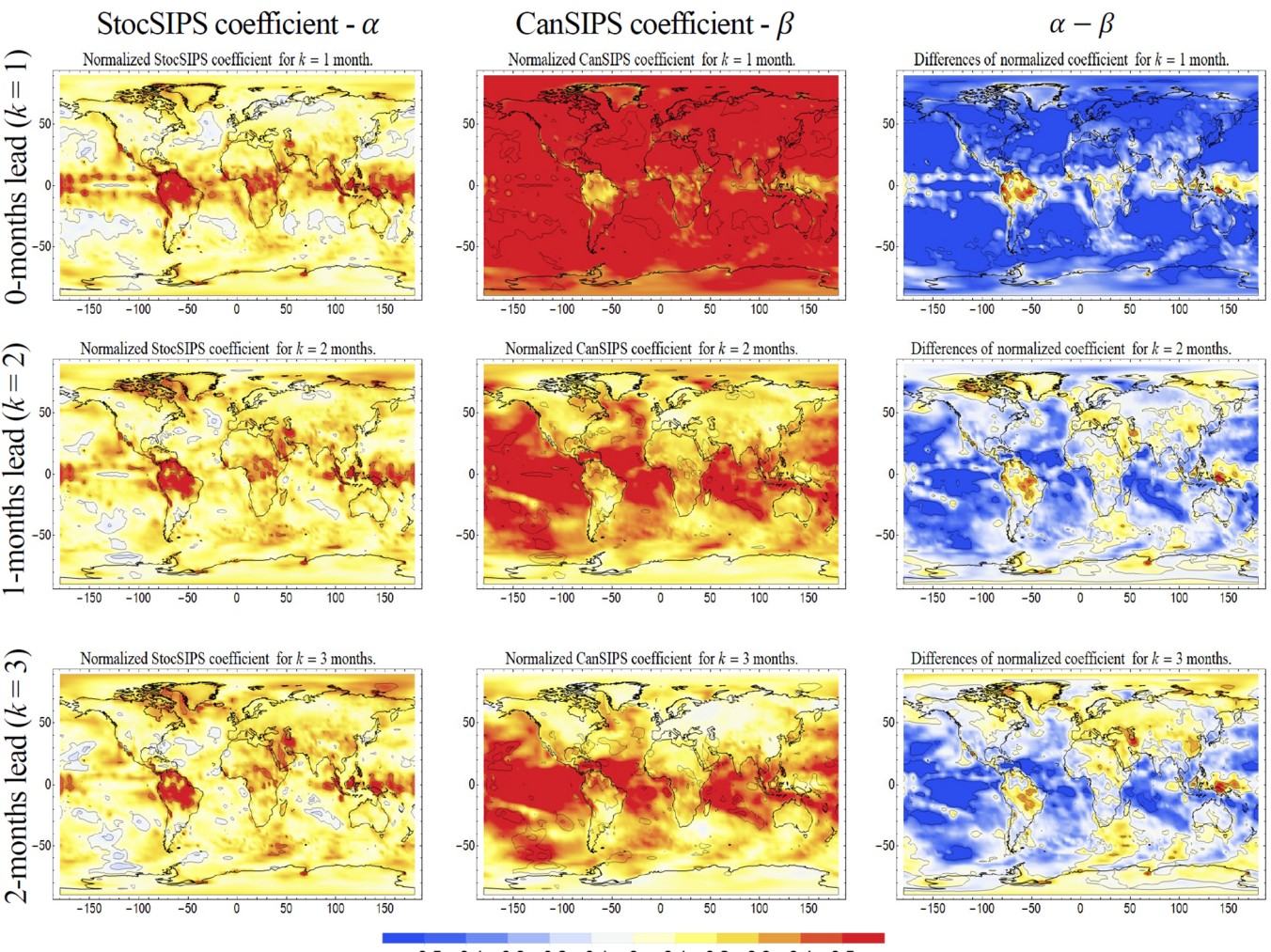

**Figure 7.** The optimum combination of copredictors gives a weight α to StocSIPS (**left column**) and β to CanSIPS (middle column, all maps at 2.5° resolution). The difference (**right column**) is red or yellow when StocSIPS gives the dominant contribution; blue or light blue when CanSIPS does. The rows (**top** to **bottom**), show how the weights evolve as the forecast lead times increase from 0 to 2 months.

The MSSS skills are shown in Figure 8. As expected, for all the lead times, CanStoc (column at the right, based on the weights from Figure 7) is the best. Also, as expected, the MSSS for all the forecasts decreases with the lead time. Notice that the CanStoc forecast at 2 months (bottom right) is a little better than the CanSIPS forecast at 1 month (middle).

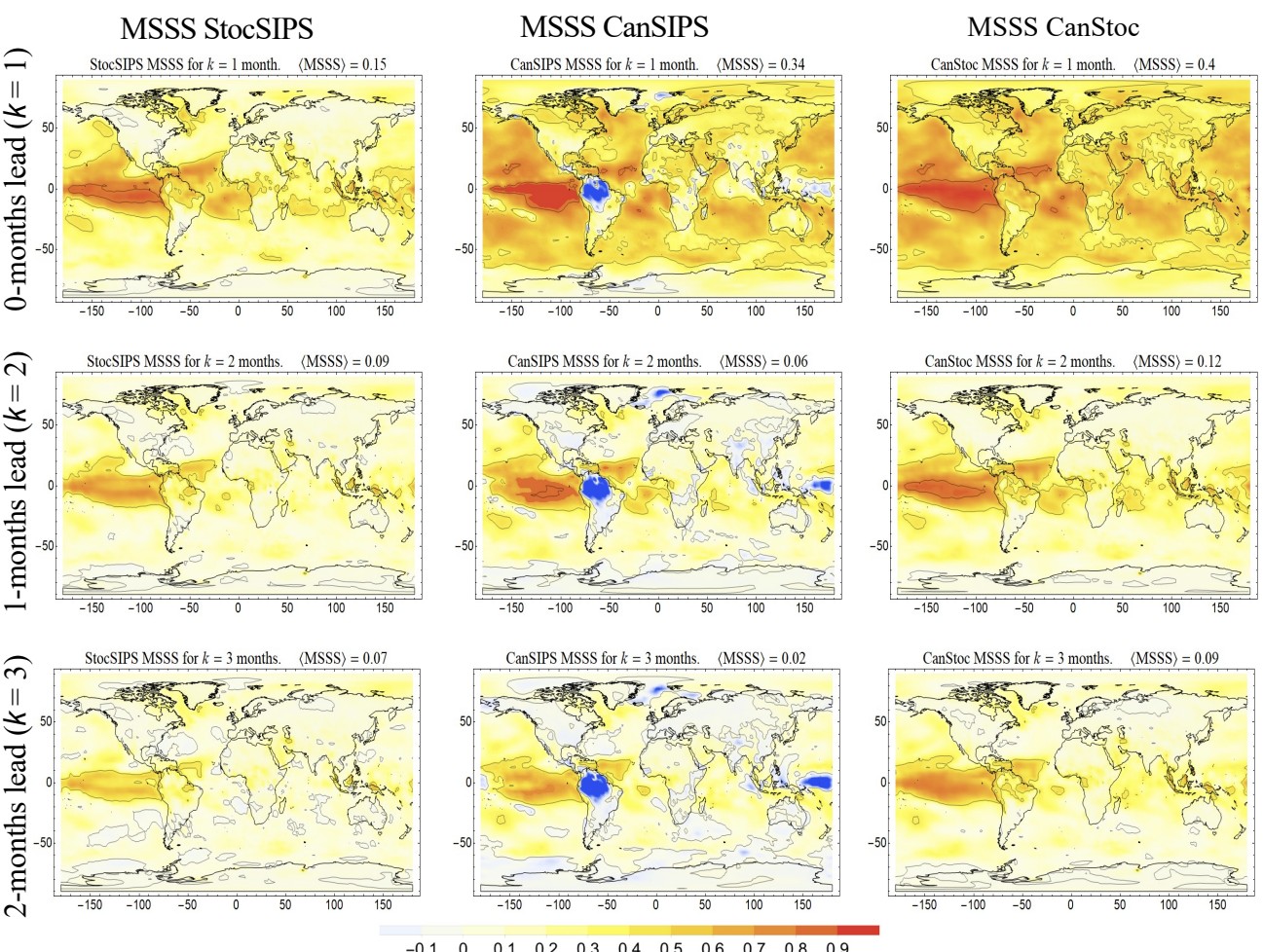

**Figure 8.** The mean square skill scores (MSSSs) for StocSIPS (**left**), CanSIPS (**middle**), and CanStoc (**right**) for 0, 1, 2 month lead times (top to bottom, all maps at 2.5o resolution). CanStoc is better (as it must be) for all lead times and, also, the MSSS decreases with the lead time. Notice that the CanStoc forecast at 2 months (**bottom right**) is a little better than the CanSIPS forecast at 1 month (**middle**).

To make the comparisons clearer, Figure 9 shows the pairwise differences in the MSSS. The left-hand "CanStoc–StocSIPS" column can be interpreted as the improvement made to StocSIPS by using CanSIPs as a copredictor. The middle column "CanStoc–CanSIPS" is the improvement provided to CanSIPS by using StocSIPS as a copredictor. Finally, the right-hand column "StocSIPS–CanSIPS" shows how much better StocSIPS is than CanSIPS (red) or worse (blue).

We see that at 0-month lead times (top row), CanSIPS is much better than StocSIPS (upper right), the main exception being over land in the tropics. This is unsurprising since CanSIPS is still within its deterministic predictability limit, and StocSIPS is only at the beginning of its range of validity. Nevertheless, when StocSIPS is used as a copredictor for CanSIPS to yield CanStoc (top middle), it still makes a significant contribution to improving the CanSIPS skill. However, already at one-month lead time (middle row), we can see that over almost all the land, StocSIPS is better than CanSIPS (middle right), but now, it is CanSIPS that allows for some further improvements in StocSIPS's skill (middle left). Finally, at the 2-month lead times (bottom row), StocSIPS has higher skill than CanSIPS, even over the majority of the ocean, and over land, CanSIPS only contributes a small amount to improving CanStoc with respect to StocSIPS.

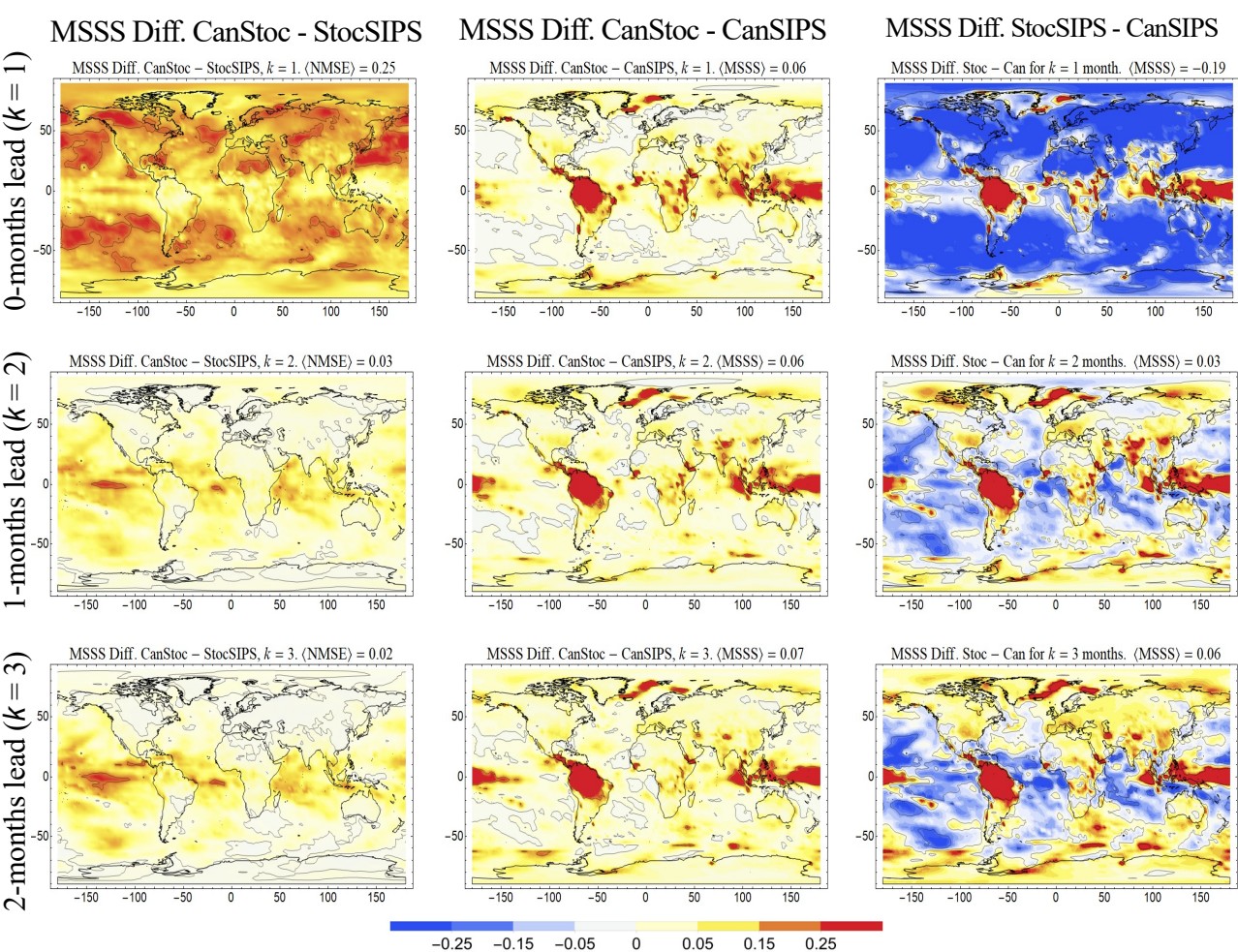

**Figure 9.** The difference in MSSS between the three pairs of three forecasts. CanStoc—StocSIPS (**left column**), CanStoc—CanSIPS (**middle column**), StocSIPS—CanSIPS (**right column**, all maps at 2.5° resolution). Note that at 0-month lead times (**top row**), CanSIPS is much better than StocSIPS (**upper right**); the main exception is over land in the tropics. Nevertheless, when StocSIPS is used as a co-predictor for CanSIPS to yield CanStoc (**top middle column**), it still leads to significant improvements in skill. At one-month lead time (**middle row**), we can already see that over almost all land, StocSIPS is better than CanSIPS (**middle right**), but that CanSIPS nevertheless allows for some further improvements in skill (**middle left**). Finally, at 2-month lead times (**bottom row**), StocSIPS has higher skill than CanSIPS even over the majority of the ocean, and, over land, CanStoc is only improved a little with respect to StocSIPS.

The most obvious feature of the maps (Figures 7–9) is that there is a systematic difference of the skills over land, ocean, midlatitudes, and tropics. To summarize this, we first show Figure 10 that summarizes the monthly resolution MSSS averaged over the globe (Figure 10a), land (Figure 10b), ocean (Figure 10c), northern midlatitudes (Figure 10d), southern midlatitudes (Figure 10e), and tropics (Figure 10f) at lead times of up to 11 months ($k = 12$). For the global average, we see that at a 1-month lead time, CanSIPS (green) is quite a bit better than StocSIPS (blue), but that even here, the co-predicted CanStoc (red) is better still. Notice that the CanSIPS MSSS rapidly decreases, becoming (nearly) zero for the lead times of 3 months and negative thereafter. A negative MSSS is primarily a consequence of poorly accounting for the anthropogenic component and annual cycle. In comparison, StocSIPS has a higher MSSS than CanSIPS for all lead times of 2 months and longer. In addition, the StocSIPS MSSS always remains positive, since it forecasts the internal and anthropogenic components separately and the latter forecast is very accurate over these time scales. For the three-month lead times and longer, CanSIPS does not add much skill to

StocSIPS, such that the CanStoc and StocSIPS MSSS are nearly the same. Interestingly, for the tropics (Figure 10f), beyond 8 months or so, neither the StocSIPS nor CanSIPS have skill on their own, but the hybrid CanStoc does have some skill.

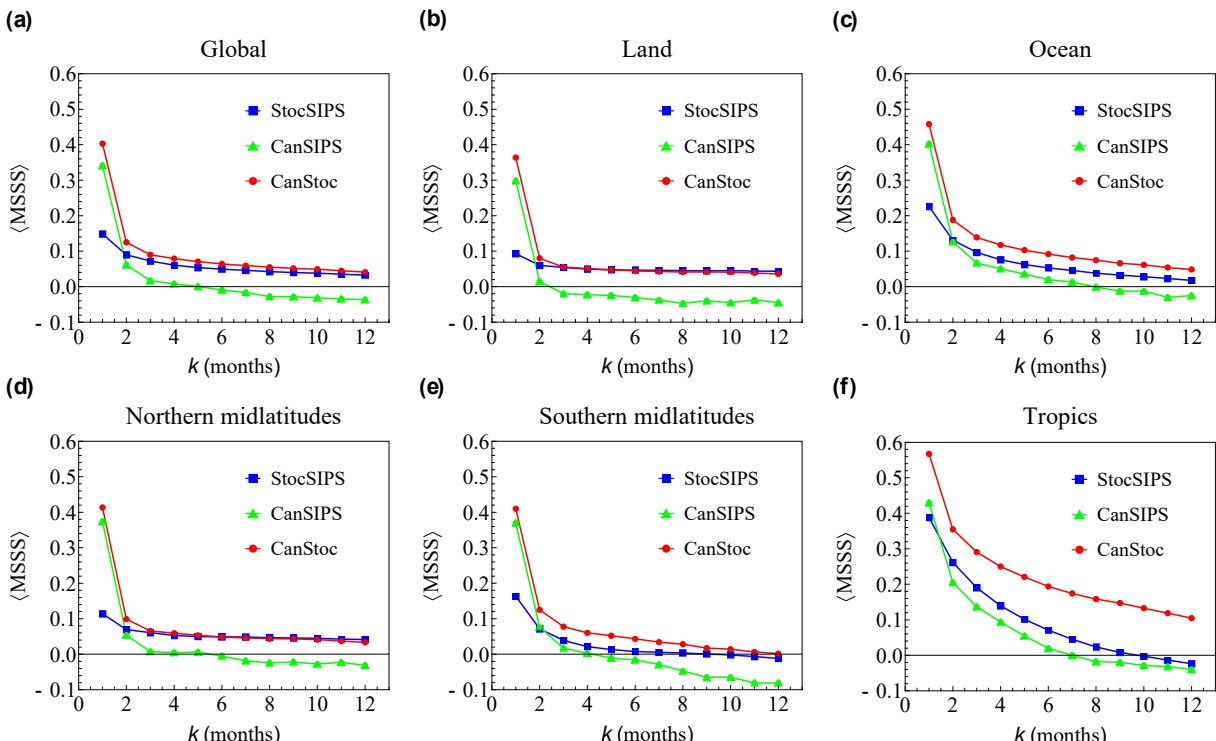

**Figure 10.** The MSSS for monthly forecasts for lead times up to 11 months ($k = 12$). The MSSS averaged over (**a**) the globe, (**b**) land, and (**c**) ocean, for lead times up to 11 months, with 2.5° resolution pixels. The bottom row are the corresponding MSSS for (**d**) northern midlatitudes (25°–67.5°), (**e**) southern midlatitudes (−25°S−−67.5°), and (**f**) tropics (−25°–25°).

If we consider the skills over land only (Figure 10b), then roughly the same comments could be made as for the globe, except that, now, CanSIPS transitions from fairly high skill at 0 month lead time ($k = 1$) to virtually zero skill at 1 month lead times ($k = 2$) after which its MSSS is negative. As a consequence, StocSIPS and CanStoc skills are nearly identical for the 1-month lead times and longer.

For the ocean pixels (right hand plot), both StocSIPS and CanSIPS show more skill than over land, with the CanSIPS MSSS staying positive until about a 6-month lead time. Nevertheless, StocSIPS has a higher MSSS for the 1-month lead times and longer. Interestingly, over the ocean, even when CanSIPs skill becomes negative, it still provides a significant boost in skill to the copredicted CanStoc (red).

Finally, instead of considering forecasts with a one-month resolution, the skills of which are averaged over all the different starting months (as in Figure 10), we may consider seasonal forecasts (Figures 11–14) with the same geographies as Figure 10. These seasonal forecasts have three-month resolutions and the forecasts start at the beginning of the season: winter (Figure 11), spring (Figure 12), summer (Figure 13), and fall (Figure 14). When compared to Figure 10, we see that both the StocSIPS and the CanStoc skills are often a little higher for the seasonal forecasts.

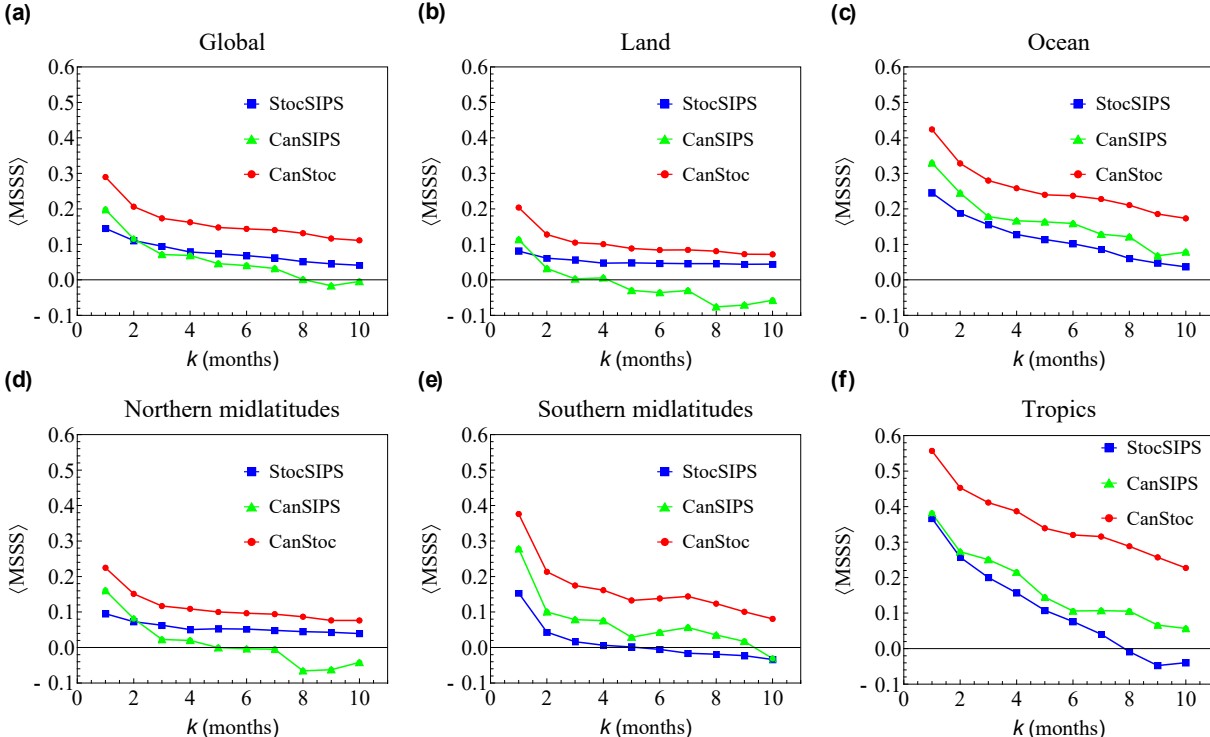

**Figure 11.** The same as Figure 10 but for seasonal winter forecasts (for the anomaly averaged over the three months: December, January, February, the forecast starting at the beginning of the season). Notice that in every case, CanStoc does significantly better than either StocSIPS or CanSIPS.

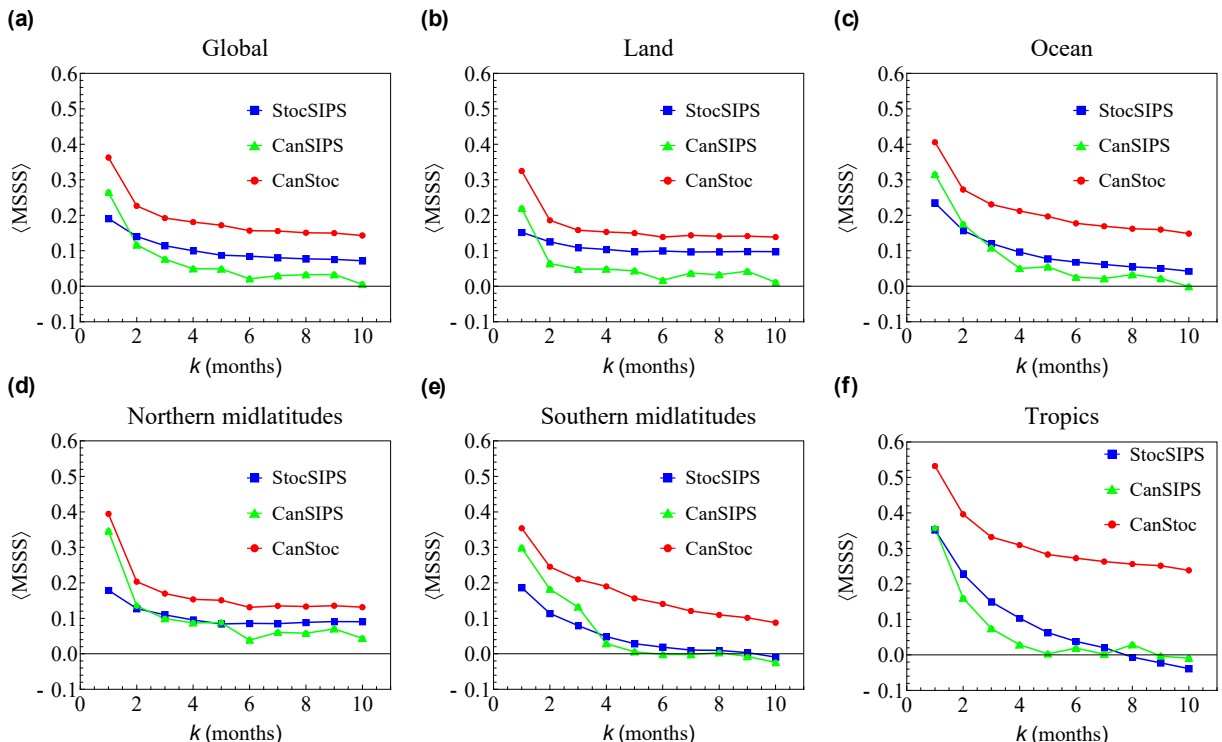

**Figure 12.** The same as Figure 11 but for the spring season (the months March, April, May). Again, notice the greatly improved CanStoc skill, especially in the tropics.

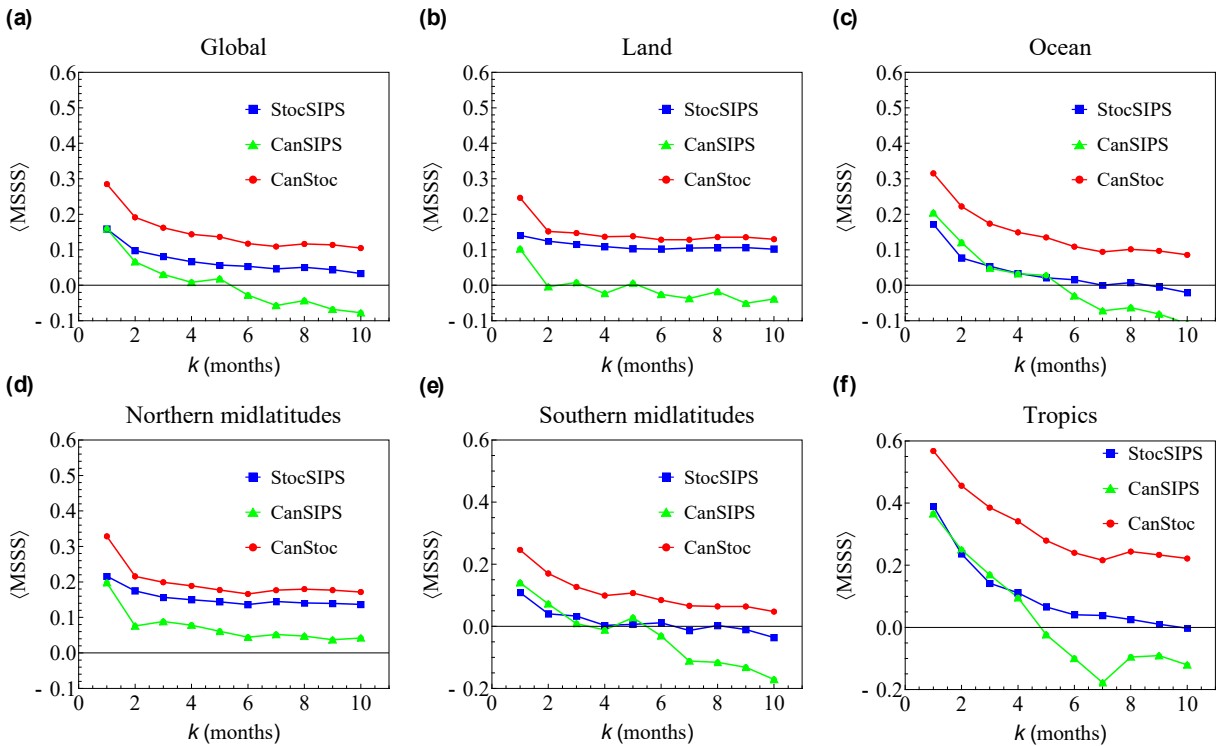

**Figure 13.** The same as Figure 11 but for the summer season (the months June, July, August).

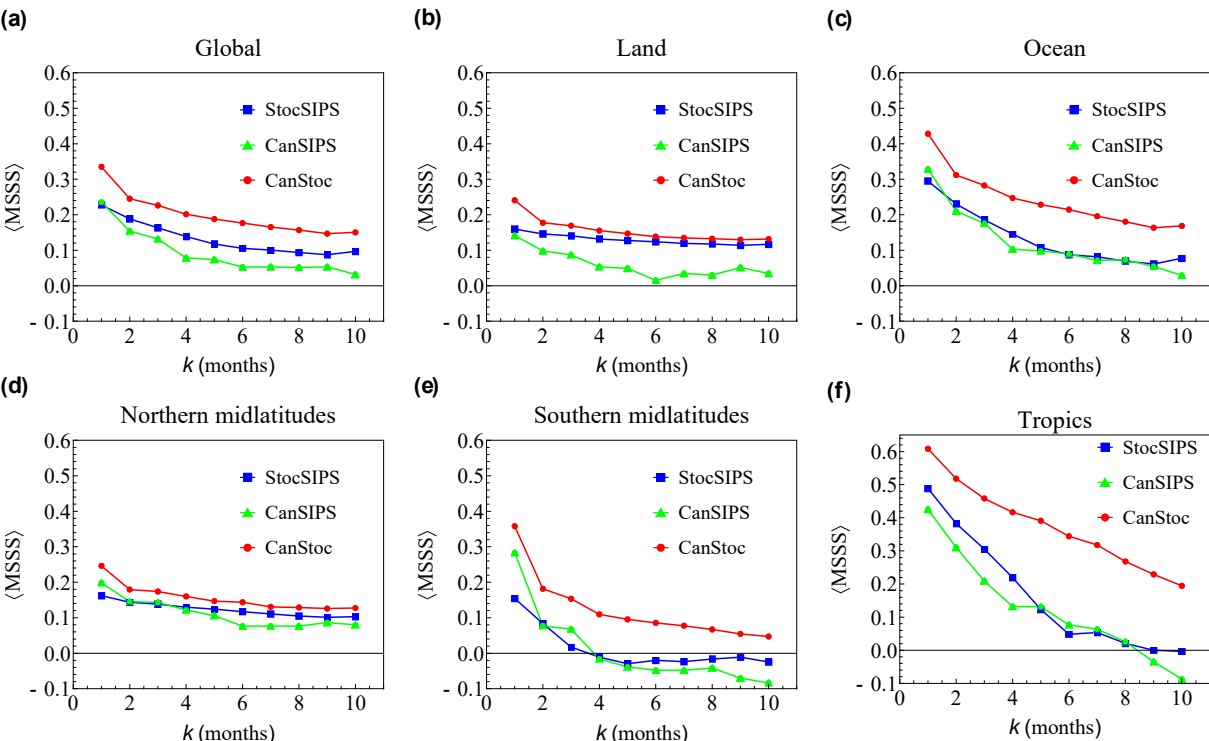

**Figure 14.** The same as Figure 11 but for the fall season (the months September, October, and November).

Although at one month, CanSIPS is better than StocSIPS, the additional information provided by StocSIPS significantly improves the hybrid CanStoc. Similarly, although for 2 months and longer, StocSIPS has higher skill, CanSIPS still contributes to an improved hybrid CanStoc, especially over the oceans. The most marked improvement is over the

tropics, where neither CanSIPS nor StocSIPS have high accuracy, but the hybrid CanStoc is much better than either. Remarkably, in the latter, for 8 months and longer, neither StocSIPS nor CanSIPS have significant skill, the CanStoc hybrid does have some.

## 3. Conclusions

For forecasts beyond the weather regime, GCMs have been optimized for either monthly, seasonal, interannual predictions (Long-Range Forecast models, LRFs) and for longer-term climate projections. Whereas the main challenge of LRF forecasting is predicting the response to (stochastic) internal forcings, the longer-term climate models attempt to average out these responses to estimate the deterministic response to anthropogenic forcings. Both variants appear to be in serious trouble. For example, while classical GCM-based LRFs can undoubtedly be further improved (for example, by using better numerical techniques such as Galerkin-like spectral atmospheric models), they have already been the subject of intense research and appear to be facing diminishing returns. Evidence for this comes from a systematic LRF model intercomparison by [36], who found disappointingly low skills for seasonal temperature forecast (especially over land), little difference in skill between the different LRFs, and little improvement when six different LRFs were combined into a single (optimum) forecast product.

In contrast, physically based stochastic models such as StocSIPS have received virtually no institutional support, and, scientifically, their development so far has only scratched the surface. The StocSIPS and CanStoc models are complementary in several ways. First, most of the CanSIPS skill comes from the weather regime (i.e., mostly the first two weeks or so) whereas the StocSIPS skill comes from the longer macroweather regime beyond the deterministic predictability limits. Second, whereas CanSIPS is a classical deterministic initial value problem, StocSIPS is a nonclassical, stochastic past value problem. It is this complementarity that already allows the hybrid CanStoc system to be attractive.

Although, as discussed, StocSIPS skill is already comparable to CanSIPS, it has much room for improvement. In particular, the current version of StocSIPS makes two simplifying approximations that will be relaxed in future versions. The first, discussed in Section 2.1, is uses the high-frequency fGn approximation to the FEBE Green's function. The theory needed to improve this already exists: fGn processes should be replaced by fractional relaxation noise (fRn) processes, as discussed in [45]. While the high-frequency behaviour is still fGn, the memory is somewhat diminished depending on the local relaxation time ($\tau$ in Equation (1)), and this, it seems, can vary enormously from place to place (there is unpublished evidence that, over some ocean regions, it can be of the order of millennia). Another FEBE improvement would be to reintroduce the neglected horizontal heat flux divergence operator $\zeta$ that couples both neighbouring and distant pixels: symbolically, in Equation (1), we make the replacement $\tau^h \partial^h / \partial t^h \to \left( \tau^{2h} \partial^{2h} / \partial t^{2h} + \zeta \right)^{1/2}$. Whereas the operator $\zeta$ is local—it describes heat fluxes between neighbouring points (pixels in discrete approximations), the square root space-time operator is nonlocal, connecting distant pixels and times. This term is part of an overall fractional space–time operator that was ignored in Equation (1), and these new nonlocal heat transport effects are currently under study. We expect that taking these into account will lead to further StocSIPS improvements.

As underlined by the recent (2021) IPCC sixth assessment report (AR6), GCMs optimized for climate projections may be in even worse straits than LRFs. Contrary to hopes, the AR6 CMIP6 climate projection uncertainties were larger than the earlier generation's CMIP5 projections (AR5, 2013), and they are now the largest ever. This "uncertainty crisis" ([19], implicitly recognized in [51]) is concisely illustrated by considering the historical evolution of the sensitivity of the global temperature to the doubling of $CO_2$. Starting with the US Academy of Sciences [52] right through to the IPCC AR6 (2021), the official 90% uncertainty limits for $CO_2$ doubling remained as a very wide range (staying close to the range 1.5–4.5 °C). Over these decades, it was hoped that, by making the models bigger and by adding more processes, the uncertainty (as quantified by the spread in the Multi Model Ensemble, MME) would be reduced. However, the AR6 shattered this hope:

while the MME spread (and hence uncertainty) *increased* from the range 1.9–4.5 °C to the range 2–5.5 °C (a 35% increase in the range), at the same time, on the contrary, the experts (based on better historical and paleo data), *reduced* their range to 2.5–4 °C (a 50% decrease in uncertainty). In effect, the experts gave the CMIP6 GCM outputs little weight. Ironically, the increased model uncertainty—the increased spread between models that compose the multi model ensemble—was a consequence of each GCM team improving its own model, effectively driving it further from the others (for discussion, see [25]).

During its historical development, the general scientific consensus for how to improve GCMs could be closely summed up as "the bigger the better", (see, e.g., [53], reiterated in [54]) with the holy grail being the attainment 1 km "cloud resolving" scales predicted for the year 2030. If bigger is no longer better, then how to move forward? Already, in the context of climate projections, [25] argued that most GCM computations were for irrelevant details that were known in advance to be wrong. Why forecast the weather in Montreal—or anywhere else—through to 31 December 2099 if the aim is to project the mean decadal 2090–2100 temperature?

An analogy may be helpful. The atmosphere is composed of molecules and, in principle, their positions and velocities could be modelled directly with particle mechanics or quantum mechanics. In comparison, GCMs are based on the equation of fluids: continuum mechanics and thermodynamics that do not even acknowledge the existence of atoms or molecules. The advantage of continuum mechanics and thermodynamics is not only that it is computationally advantageous when compared to tracking and modelling all the molecules. The real advantage is that it ignores the *irrelevant details*. However, the atmosphere is not a single vortex or even a small collection of vortices, it is composed of a huge number of (turbulent) vortices. We should therefore aim to develop a higher level theory that ignores the irrelevant weather scale details, a theory and model based on the higher level statistical laws that result from the collective interactions of huge numbers of clouds, eddies, structures.

The candidate high-level theories should, from the outset, be stochastic and they should start with the weather—macroweather transition scale of about 10 days [25]. In order to handle the large range of scales, they should exploit a basic symmetry of the governing equations that is well obeyed by the models and the real world: the statistical spatial and temporal scaling of the atmospheric fields. Finally, they should also respect energy balance (and probably other conservation laws, symmetries). Ref. [25] further argued that the Fractional Energy Balance Equation (FEBE) was a promising candidate for such a macroweather and climate model. The FEBE is a consequence of improving the (highly successful) Budyko–Sellers-type energy balance models. Ref. [24] showed how the FEBE could be used to make climate projections through to the year 2100 that had about half the uncertainty of the current GCM Multi Model Ensemble (MME). The FEBE median projection with its 90% uncertainty limits was almost completely contained within the corresponding CMIP5 and CMIP6 projection uncertainties, such that the two approaches completely agreed with each other. The fact that two qualitatively different model types give essentially the same median projections gives strong support to both model approaches. Finally—just as we show here using LRF models—[25] showed how such stochastic FEBE based projections can be combined with GCM projections to yield even better "hybrid" projections.

Hybrids obtained by combining GCMs with stochastic models, whether for climate projections or—as here—for LRFs, are particularly promising, because the source of skill in both cases is quite different (see [25] for such a hybrid). Whereas the GCMs are classical initial value problems that effectively only use spatial information (the initialization fields at $t = 0$), on the contrary, the StocSIPS stochastic LRF is a *past value* problem that only uses past data: if the data series are long enough, then adding spatial correlation information (such as from teleconnections) makes no improvement since other pixels have no Granger causality [30]. As we show, using the Canadian Seasonal and Interannual Prediction system (CanSIPS) and the FEBE-based StocSIPS model as linear copredictors—the CanStoc hybrid

system—the two model types are indeed highly complementary. At zero-month lead times, CanSIPS has an overall (globally averaged) higher skill (MSSS = 0.35) than StocSIPS (especially over oceans), (MSSS = 0.17), but nevertheless, the information from StocSIPS significantly improves CanSIPS, such that the CanStoc MSSS = 0.41 (see Figure 10a–c). Conversely, for one-month and longer lead times, StocSIPS has generally higher skill (MSSS = 0.09) than CanSIPS (MSSS = 0.06), but CanSIPS nevertheless contributes to the optimal CanStoc forecast (MSSS = 0.13). CanStoc's skill was particularly impressive over the tropics: even at 8-month lead times and longer, when both CanSIPS and StocSIPS had negligible skill (MSSS $\approx$ 0 to $-0.05$), the hybrid combination (CanStoc) had some skill (MSSS $\approx$ 0.14, Figure 10f). Similarly, CanStoc did significantly better in the three-month resolution, and the seasonal forecasts are shown in Figures 11–14.

Interestingly, LRFs currently place huge effort on postprocessing to correct for known biases (e.g., [39]). In this regard, StocSIPS could be regarded as an optimal post-processing method. Not only does it demonstrably lead to improved forecasts, but it is physically based in the FEBE; it is a not "black box"-type statistical postprocessing "massage". The reason for this is their complementarity: GCMs mostly have deterministic skill and this is mostly over the first two weeks, such that their monthly average skills are already somewhat low. In comparison, StocSIPS only works for times longer than about two weeks, therefore, the one-month lead time is somewhat short. It is therefore logical to conclude that StocSIPS provides a bias correction in the GCMs at one-month lead times, and that CanSIPS provides a bias correction to StocSIPS at the longer lead times.

**Author Contributions:** Conceptualization, S.L. and L.D.R.A.; methodology, S.L. and L.D.R.A.; software, L.D.R.A.; validation, L.D.R.A.; formal analysis, L.D.R.A.; investigation, L.D.R.A.; resources, L.D.R.A.; data curation, L.D.R.A.; writing—original draft preparation, S.L. and L.D.R.A.; writing— review and editing, S.L.; visualisation, L.D.R.A.; supervision, S.L.; project administration, S.L.; funding acquisition, S.L. All authors have read and agreed to the published version of the manuscript.

**Funding:** This research was funded by McGill University, Fessenden professor (Lovejoy), and National Science and Engineering Research grant number 222858.

**Data Availability Statement:** The CanSIPS data in GRIB2 format may be accessed at https://dd. weather.gc.ca/ensemble/cansips/grib2/hindcast/raw/ (accessed on 20 April 2021). The reference observational datasets are monthly average surface temperature (T2m) from the European Centre for Medium-Range Weather Forecasts (ECMWF) ERA-Interim Reanalysis https://www.ecmwf.int/en/ forecasts/dataset/ecmwf-reanalysis-interim. The data were accessed 20 April 2021 and cover the period January 1979 to August 2019. All data were interpolated to a 2.5° latitude × 2.5° longitude grid across the globe for a total of 73 × 144 = 10,512 grid points.

**Acknowledgments:** We thank Dave Clarke and Roman Procyk for their helpful discussions and comments.

**Conflicts of Interest:** The authors declare no conflict of interest.

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
