# Peer review of "CanStoc: A Hybrid Stochastic–GCM System for Monthly, Seasonal and Interannual Predictions"

_2674-0494, doi:10.3390/meteorology2040029_

Round 1

Reviewer 1 Report (Previous Reviewer 1)

Comments and Suggestions for Authors

CanStoc: A Hybrid Stochastic – GCM System for Monthly, Seasonal and Interannual Predictions

Authors:  S Lovejoy and L D R Amador

Review of Meteorology 2531195 article REVISED

Recommendation: Publish after minor revision

Summary:

As in my previous review: The authors combine the results from ensembles of General Circulation Models (GCMs) and empirical stochastic models that are fitted to the global climate to obtain multi month temperature forecasts. The empirical stochastic models remove some of the climate drift biases of the GCMs particularly in the tropics. This improves the skill slightly.

General Comments:

The paper is considerably improved on the first attempt. There are however some minor issues that should be corrected before publication.

Specific Comments:

Title: Use MDPI convention (see above)

L92: so So

L100:  Make link into a reference.

L140-143: Awkward – rephrase.

Equation (1): State that you use the standard gamma function. Using p as a dummy variable is awkward – t’ or sigma might be better. T’ is presumably delT/delp → delT/delt’ in changed notation. T → T(lambda, thi, t) ???

L247: 5.5C CO2 5.5C/CO2 ???

L529: Continue onto next line.

References: Correct variable size fonts. Set references in MDPI convention.

Author Response

Recommendation: Publish after minor revision

Summary:

As in my previous review: The authors combine the results from ensembles of General Circulation Models (GCMs) and empirical stochastic models that are fitted to the global climate to obtain multi month temperature forecasts. The empirical stochastic models remove some of the climate drift biases of the GCMs particularly in the tropics. This improves the skill slightly.

General Comments:

The paper is considerably improved on the first attempt. There are however some minor issues that should be corrected before publication.

Au: Thanks for the positive evaluation!

Specific Comments:

Title: Use MDPI convention (see above)

L92: so → So

Au: Thanks.

L100:  Make link into a reference.

Au: Thanks.

L140-143: Awkward – rephrase.

Au: Thanks.

Equation (1): State that you use the standard gamma function. Using p as a dummy variable is awkward – t’ or sigma might be better. T’ is presumably delT/delp → delT/delt’ in changed notation. T → T(lambda, thi, t) ???

Au: We added the comment about the Gamma function, but we left the “p” as dummy.  If instead we use t’, this is reminiscent of the derivative T’ that is also used.  Use of primes with two different meanings in the same integrand will be confusing.

L247: 5.5C CO2 → 5.5C/CO2 ???

Au: Thanks.

L529: Continue onto next line.

Au: We didn’t understand.

References: Correct variable size fonts. Set references in MDPI convention.

Au: Thanks.

Reviewer 2 Report (New Reviewer)

Comments and Suggestions for Authors

The concept of macro-weather is an interesting and novel idea as previously proposed by the authors. I totally agree with the authors that “Why forecast the weather in Montreal - or anywhere else - through to Dec. 31st, 2099 if the aim is to project the mean decadal 2090-2100 temperature?”.

In this new study, the authors further put this concept into practical application in a hybrid way by combing physics-based model and stochastic energy-balanced model for longer-range forecasts. The preliminary results look promising. Therefore, I suggest publishing it for further discussion and development.

Lines 50-51: A typo. Please correct this: “(but stochastic but not fractional)”

Author Response

The concept of macro-weather is an interesting and novel idea as previously proposed by the authors. I totally agree with the authors that “Why forecast the weather in Montreal - or anywhere else - through to Dec. 31st, 2099 if the aim is to project the mean decadal 2090-2100 temperature?”.

In this new study, the authors further put this concept into practical application in a hybrid way by combing physics-based model and stochastic energy-balanced model for longer-range forecasts. The preliminary results look promising. Therefore, I suggest publishing it for further discussion and development.

Lines 50-51: A typo. Please correct this: “(but stochastic but not fractional)”

Au: Thanks!

Reviewer 3 Report (New Reviewer)

Comments and Suggestions for Authors

Comments on the Quality of English Language

The quality of English language is good.

Author Response

See the attached pdf.

Round 2

Reviewer 3 Report (New Reviewer)

Comments and Suggestions for Authors

The detail included in this revised manuscript greatly enhances the message the authors wish to convey.

This manuscript is a resubmission of an earlier submission. The following is a list of the peer review reports and author responses from that submission.

Round 1

Reviewer 1 Report

Comments and Suggestions for Authors

See attachment

Comments on the Quality of English Language

The article has the feel of a first draft that has not been proof-read by the authors. Too much hype and not enough scientific detail for the reader to gain something from the paper.

Reviewer 2 Report

Comments and Suggestions for Authors

If this paper is written to propose a new method for monthly, seasonal and interannual predictions, please stick to it and refrain from passing judgement on GCMs general ability to produce valid and accurate climate predictions over time horizons exceeding the purpose of this paper. Hence please rewrite this paper to make this point clear.

Comments on the Quality of English Language

Not necessary.

Reviewer 3 Report

Comments and Suggestions for Authors

This paper attempt to present a new research on how to combine StocSIPS with a classical coupled GCM system (CanSIPS) into a hybrid system (CanStoc) whose skill is better than either. However, it is quite difficult to find out the merit and highlight of this study. The objective and methodology is poorly presented. The design of the numerical experiment and the result is also not adequately introduced. The findings of the research is not quite clear with very weak support and analysis from the datasets the authors had provided.   

Comments on the Quality of English Language

Not quite good.
